# Design and Characterization of a Sharp GaAs/Zn(Mn)Se Heterovalent Interface: A Sub-Nanometer Scale View

**DOI:** 10.3390/nano10071315

**Published:** 2020-07-04

**Authors:** Davide F. Grossi, Sebastian Koelling, Pavel A. Yunin, Paul M. Koenraad, Grigory V. Klimko, Sergey V. Sorokin, Mikhail N. Drozdov, Sergey V. Ivanov, Alexey A. Toropov, Andrei Y. Silov

**Affiliations:** 1Department of Applied Physics and Institute for Photonic Integration, Eindhoven University of Technology, P.O. Box 513, 5600 MB Eindhoven, The Netherlands; s.koelling@tue.nl (S.K.); P.M.Koenraad@tue.nl (P.M.K.); 2Institute for Physics of Microstructures RAS, 603950 Nizhny Novgorod, Russia; yunin@ipmras.ru (P.A.Y.); drm@ipmras.ru (M.N.D.); 3Lobachevsky State University of Nizhny Novgorod, 603950 Nizhny Novgorod, Russia; 4Ioffe Institute, 194021 St. Petersburg, Russia; gklimko@mail.ru (G.V.K.); sorokin@beam.ioffe.ru (S.V.S.); ivan@beam.ioffe.ru (S.V.I.); toropov@beam.ioffe.ru (A.A.T.)

**Keywords:** diluted magnetic semiconductors, diffusion, ZnSe/GaAs interface, Mn impurities, dopants, segregation

## Abstract

The distribution of magnetic impurities (Mn) across a GaAs/Zn(Mn)Se heterovalent interface is investigated combining three experimental techniques: Cross-Section Scanning Tunnel Microscopy (X-STM), Atom Probe Tomography (APT), and Secondary Ions Mass Spectroscopy (SIMS). This unique combination allowed us to probe the Mn distribution with excellent sensitivity and sub-nanometer resolution. Our results show that the diffusion of Mn impurities in GaAs is strongly suppressed; conversely, Mn atoms are subject to a substantial redistribution in the ZnSe layer, which is affected by the growth conditions and the presence of an annealing step. These results show that it is possible to fabricate a sharp interface between a magnetic semiconductor (Zn(Mn)Se) and high quality GaAs, with low dopant concentration and good optical properties.

## 1. Introduction

Diluted Magnetic Semiconductors (DMS) are an interesting class of materials, obtained by adding a small concentration of magnetic dopants in a semiconductor (e.g., Ga(Mn)As) [1]. These materials have many potential applications in the field of spintronics, including devices with high speed and low power consumption [2], magneto-optical devices [3], and semiconductor based spin valves [4,5]. One of the challenges in this field is to create DMS with strong magnetic response (i.e., showing ferromagnetic [6] or superparamegnetic [7] properties), without compromising the opto-electronic properties of the semiconductor. The magnetic properties of DMS can be enhanced by growing heavily doped semiconductors at low temperature [8], but with this approach the optical properties of the semiconductor are quenched by the high concentration of Mn dopants and intrinsic defects, introduced by the growth at low temperature.

A different strategy proposed to address this problem is to create a sample with two neighboring, distinct regions: one with a high Mn concentration and good magnetic properties, the other retaining a low dopant concentration and good opto-electronic properties [9,10]. Heterostructures of III–V/II–VI semiconductor materials are a promising system to fabricate such structures [11]. Semiconductor Quantum Wells (QWs) can be realized exploiting the difference in band offset while using a combination of nearly lattice-matched semiconductors, such as GaAs and ZnSe (the difference in lattice parameters is less than 0.3%). Two approaches have been envisioned: the first involves the creation of a GaAs/ZnSe QW where Mn is introduced in the II–VI part of the stack. The carriers confined in the GaAs QW interact via exchange interaction with ${\mathrm{Mn}}^{2+}$ ions from the ZnSe barrier, gaining a net magnetization [9,12,13]. The second approach is based on the fabrication of a high quality GaAs layer, with a MnSe delta layer on top, followed by a ZnSe barrier. In this scenario, the back diffusion of Mn in GaAs during a post growth annealing step can be exploited to create a high quality DMS, with a small amount of intrinsic defects. This material is expected to have a higher Curie temperature with respect to low temperature grown Ga(Mn)As, where the high concentration of intrinsic defects has a detrimental effect on the $T_{C}$ [14,15].

The behavior of Mn atoms at the III–V/II–VI interface is of great importance to determine the most favorable approach. If Mn impurities can diffuse extensively, the III–V/II–VI interface can be exploited to realize high quality DMS; conversely if the diffusion length is small or negligible across the interface, the creation of defect free III–V/II–VI QWs appears more promising.

In this work, we thoroughly investigate the Mn diffusion profile across a GaAs/MnSe/ZnSe heterostructure combining three different experimental techniques: Secondary Ion Mass Spectroscopy (SIMS), Cross Sectional Scanning Tunneling Microscopy (X-STM), and Atom Probe Tomography (APT). This unique combination allowed us to overcome the limitations of each technique, collecting data on the Mn distribution with a wide dynamical range and a high spatial resolution. This information allowed us to gain a comprehensive understanding of the diffusive behavior of Mn impurities in this system.

## 2. Materials and Methods

In this work, three samples (*a*, *b*, *c*) have been investigated, with differences in the stack structure around the III–V/II–VI interface. The samples are all composed of a GaAs buffer layer and a thin MnSe layer, followed by a ZnSe layer. All samples are grown following a similar strategy for heterovalent interface formation. First, a GaAs buffer layer is grown by MBE on p^+^-GaAs substrates in a III–V chamber at 580 °C. Then, the sample is cooled whilst preserving a (2 × 4) As surface reconstruction and is transferred to a II–VI MBE growth chamber. In samples *b* and *c*, the buffer layer has a thickness of 100 nm, while, in sample *a*, it is optimized for STM analysis, and the buffer thickness is 5 nm.

In the II–VI chamber, the GaAs surface is heated up to 300 °C and two approaches are followed for the growth of MnSe. In sample *a*, conventional MBE growth is performed, allowing both Mn and Se in the chamber simultaneously (Se/Mn > 1). A layer 5 ML thick is grown in this way. In samples *b* and *c*, the MnSe layer is grown in a Migration Enhanced Epitaxy (MEE) mode, performing three cycles of alternate deposition of Mn and Se with an intended thickness of 3 ML. After this stage, sample *c* is overgrown with 3 ML (1 nm) of ZnSe cap layer, and it is subsequently annealed in situ for one hour at 600 °C under the Se flux to enhance Mn diffusion into GaAs. Finally, all the structures are capped with a 70 nm-thick ZnSe layer, maintaining the growth temperature at 300 °C and the ratio Se/Zn≥1, close to unitary stoichiometric conditions. A graphical overview for the samples is shown in the Appendix A.

It should be noted that, in the samples grown by MEE (*b* and *c*), the real thickness of the MnSe layer can be less than the intended one by a factor of 2 (i.e., 1.5 ML = 0.5 nm rather than 1 nm) [16], similarly to the case of MEE growth of Zn(Cd)Se at 300 °C, where at maximum 0.5 ML, Zn(Cd)Se is deposited per one MEE cycle due to the existence of a stable c(2×2)-Zn(Cd) surface reconstruction. It should be noted that basic experiments on MEE of MnSe, to the best of our knowledge, have not been reported so far.

The three samples are studied with SIMS, APT, and X-STM to obtain a comprehensive picture of the Mn impurities’ diffusive behavior. Further details on these three experimental techniques are provided in the Appendix A.

By X-STM measurements, we analyzed the cross section of the semiconductor stack with atomic resolution, identifying defects and single dopants. Due to the presence of extended defects and the low conductivity in proximity of the III–V/II–VI interface, we used X-STM only to study the concentration of Mn impurities in the III–V part of the stack. SIMS measurements are performed using a probing beam of Bi^+^ ions, probing the front tail of the elements distribution with a resolution of 2–3 nm and a sensitivity in the range of 10 ppm. APT measurements are performed using a laser to enable field emission of ions. The collected ions are analyzed using a mass spectrometer and, in order to maximize the accuracy of the spatial distribution of the elements, only the main isotope for each element is considered in the data analysis. The depth resolution achieved in this measurement with this technique is about 0.2 nm, as discussed in Appendix A [17,18,19,20].

## 3. Results

The results for the APT measurements on sample *b* are shown in Figure 1. The concentration profiles are fitted using an error function fit, according to a simple diffusion model. In particular, the Mn profile can be described with a combination of two error functions, which describe the diffusion at the front tail and at the back tail of the MnSe thin film. Therefore, for this concentration profile, the fitting function is given by:(1)$$Mn\left(x\right)=\frac{C}{2}\left[erf\left(\frac{x-x_{1}}{2\lambda }\right)-erf\left(\frac{x-x_{2}}{2\lambda }\right)\right]$$
where *C* is the Mn concentration in the MnSe layer, $x_{1}$ and $x_{2}$ define the thickness of the layer, and λ is the diffusion length.

The fitting parameters obtained for the different elements are summarized in Table 1. The total depth resolution of the APT measurements is about 0.2 nm [17,18,19,20]; therefore, the concentration profiles are not limited by the instrument response function. Since the primary goal of this study is to investigate diffusion and intermixing phenomena taking place at the interface, we chose to neglect the contribution of As clusters to the total As count. This approach allows us to be more sensitive to As inside ZnSe and follow the diffusion profile more accurately, but an apparent difference in concentration between Ga and As in the APT results is observed in Figure 1. Further details on this phenomenon are described in Appendix A.

These results show that the MnSe layer is heavily intermixed during growth, being Mn_*x*_Zn_(1−*x*)_Se with x=0.24. The layer is about 3 nm thick, which is in qualitative agreement with the amount of MnSe introduced in the chamber, since a pure MnSe layer about 1.0 nm thick was expected. Mn segregation during ZnSe growth is a very probable mechanism for MnSe redistribution into the MnZnSe alloy, as the Mn–Se bond energy is smaller than that of Zn–Se [21].

The fitting model is based on Equation (Equation 1) and only takes Mn diffusion into account. The fits are in good agreement with the experimental profiles, over the concentration range detectable by APT on this system. Some deviations are observed for the tails of the Mn and the Ga profile in the ZnSe layer, where they appear to be slightly more extended. This asymmetric distribution of Mn atoms likely originates from preferential Ga(Mn)–Se bonds formation with respect to Zn(Mn)–As. This is due to efficient As substitution by Se and to an enhancement of Ga diffusivity into ZnSe, which was previously observed from XPS and SIMS measurements on a GaAs/ZnSe interface fabricated in the same growth regime [22]. Moreover, segregation effects, not captured by the previous fitting model, may also contribute to this discrepancy.

For both interfaces, the APT data show that the Mn concentration is falling by at least two orders of magnitude within 1 nm. The noise level in the APT measurements is around 0.1% for Mn, which allows the study of the species’ concentration over two orders of magnitude. However the limited sensitivity of these measurements prevents an in-depth analysis of the tail of the profile.

In order to probe the concentration of Mn atoms in the III–V semiconductor with a greater sensitivity, we perform X-STM measurements. An area of about 1 μm long, close to the MnSe/GaAs interface, has been analyzed. The surface obtained upon cleaving is often rough and non-conducting in proximity of the GaAs/ZnSe interface. Through a careful optimization of the scanning parameters, we have been able to perform STM analysis in close proximity to the interface: in sample *b* and *c*, the area analyzed is about 10 nm away from the MnSe/GaAs interface. In sample *a*, which has been carefully engineered to enable STM measurements as close as possible to the interface, images could be obtained at less than 5 nm from the interface. The large portion of sample analyzed allow us to detect a Mn concentration as low as $10^{15}\;{\mathrm{atoms}/\mathrm{cm}}^{3}$.

The results for sample *a* are shown in Figure 2a; several Be atoms in GaAs are identified, thanks to their characteristic triangular shape when imaged under empty states imaging condition [23]. From these measurements, a Be concentration of $6\cdot 10^{18}\;{\mathrm{atoms}/\mathrm{cm}}^{3}$ can be estimated, which is in good agreement with the nominal concentration expected for this layer ($2.5\cdot 10^{18}\;{\mathrm{atoms}/\mathrm{cm}}^{3}$). The Be concentration decreases, as expected, towards the II–VI interface. In the thin intrinsic GaAs layer, some Be atoms are detected, but different features are also observed, which show a bow-tie shape similar to the contrast induced by Mn atoms [24]. The estimated concentration of Mn atoms in the thin region next to the GaAs/MnSe interface is about $7\cdot 10^{17}\;{\mathrm{atoms}/\mathrm{cm}}^{3}$. This observation is compatible with the result of the APT measurement on sample *b*, shown in Figure 1, where a rapid decay of the Mn concentration in GaAs is observed. Within 2 nm from the MnSe interface, the Mn concentration decreases of two order of magnitudes, falling below the noise level ($10^{19}\;{\mathrm{atoms}/\mathrm{cm}}^{3}$).

In (b), an STM image of the sample *b* is shown. The presence of a more extended intrinsic layer enable us to spatially distinguish the Mn atoms from the Be atoms more accurately, but it reduces the conductivity in proximity of the II–VI interface. In the region analyzed, which is approximately 10 nm from the expected position of the interface, no Mn atoms were detected. This results in an upper limit for the Mn concentration in this area of $1\cdot 10^{15}\;{\mathrm{atoms}/\mathrm{cm}}^{3}$.

In both STM images of Figure 2, an irregular wave-like pattern is observed towards the top of the images. This feature is likely caused by irregularity in the electronic contrast near the III–V/II–VI interface. Steps and defects created on the [110] GaAs surface during cleavage are known to locally modify the electronic properties of the surface [25], introducing localized states and unscreened long range Coulomb potentials. Therefore, the presence of this wave-like feature confirm that the images are taken in close proximity of the interface.

More detailed information regarding the Mn distribution in the ZnSe is obtained from SIMS measurements. SIMS has a wide dynamical range, which spans four orders of magnitude and allows a thorough study of the concentration profile’s tails. However, the profile obtained by SIMS may be affected by artifacts due to the back-scattering of ions from different depths in the sample, especially in the back tail of a concentration profile, where the concentration of a specific element decreases. The absence of Mn, Zn, and Se in GaAs is confirmed by both APT and STM measurements and, for this reason, we decided to neglect the back tail of the concentration profile measured by SIMS. In our work, the same fit functions used for the APT measurements are employed to fit the front tail of three different profiles: Mn, Ga, and As. In the case of the Mn profile, a Lorentzian distribution is added to describe the front tail of the Mn peak. The result of SIMS measurements on sample *b* is shown in Figure 3.

The diffusion length estimated from the SIMS measurements is about 1 nm longer than the data obtain from the APT analysis. This suggest that the lower vertical resolution of the SIMS causes a broadening of the profile of about 1 nm. The diffusion length for Mn in different samples are reported in Table 2 The SIMS measurements confirm the presence of an extended tail in the growth direction for both the Mn and the Ga profiles.

The Mn profiles obtained from SIMS measurements on all the samples are shown in Figure 3b. The STM measurements have shown that the diffusion of Mn in GaAs, is limited within a few nanometers from the interface; therefore, the back tail of the SIMS data will not be considered.

The Mn profile’s front tail can be divided into two components: a diffusive component, which decays rapidly in the growth direction, and one slowly decaying contribution, probably originated from Mn segregation. Since these two phenomena occur on very different length scales, on a first approximation, we can treat them separately. Under this assumption, the Mn profile can be described as $Mn\left(x\right)=Mn_{diff}\left(x\right)+Mn_{segr}\left(x\right)$.

The diffusive component will be described with the same fitting model used for the APT data (Equation (Equation 1)), allowing the parameters *C*, $x_{1}$, λ to vary. In order to fit, only the front tail the fit is limited in the range $x=\left[x_{1},\left(x_{1}+\Delta x\right)\right]$, where Δx is the thickness of the MnSe layer obtained from the APT profile. The slowly decaying component of the Mn profile is fitted using a Lorentzian distribution. The function used for this second fitting is given by:(2)$$Mn_{Tail}\left(x\right)=\frac{C_{L}}{\pi \lambda_{L}}\left[\frac{1}{\pi }\frac{\lambda_{L}}{{\left(x-x_{c}\right)}^{2}+\lambda_{L}^{2}}\right]$$
where $\lambda_{L}$ is the Full Width Half Maximum and $C_{L}$ is the Mn concentration at the maximum of the Lorentzian distribution. This value provides qualitative information on the ratio of Mn atoms involved in the segregation process. The center of the Lorentzian distribution is fixed in the center of the MnSe layer, estimated from the fit of the diffusive component. Since this distribution should describe only the slowly decreasing component of the Mn profile, the fitting is performed in a region where the diffusive contribution is negligible $\left[0,x_{1}-4\lambda \right]$. Here, λ is the diffusion length extrapolated from the fit of the diffusive component, performed in the first step. The combination of the diffusive and segregation components describes the whole Mn profile, as shown in Figure 3b. The main fitting parameters for the three samples are summarized in Table 2. In the last column, the ratio between Mn involved in segregation and Mn involved in diffusive processes is reported.

The comparison of the fit parameters reported in Table 2 for sample *a* and *b* shows the differences in the Mn distribution due to different growth methods (MBE and MEE, respectively). In particular, a strong discrepancy between the two samples is observed in the ratio $C_{L}/C$, indicating that a larger fraction of Mn atoms is involved in the segregation process for sample *a*, grown by MBE. It should be noted that the MBE and MEE growth are performed under similar conditions, as described in the experimental section. The comparison between sample *b* and *c* shows the effect of annealing on the Mn distribution: in sample *c*, which was annealed, a larger fraction of Mn atoms participate to segregation, as indicated by the ratio $C_{L}/C$. During the growth of sample *c*, the annealing occurs after the deposition of a 3 ML ZnSe cap, while the rest of the ZnSe layer is grown under nominally normal temperature.

We argue that the differences in segregation behavior can be explained by two factors:The different bond strengths of Zn and Mn with Se and the interdiffusion taking place during growth and annealing.The Zn–Se bond is stronger than the MnSe bond; therefore, when the Se concentration at the growth surface is sufficiently high, there is no competition between Zn and Mn to form a bond with Se, and the segregation of Mn is suppressed [21].

MBE growth (as in sample *a*) causes an Se deficiency at the surface, with respect to sample *b*, grown by MEE [16]. This induces the substitution of Mn atoms, already incorporated in the surface, by Zn atoms, explaining the stronger Mn segregation observed in sample *a* with respect to sample *b*. In case of sample *c*, the annealing step enhances the interdiffusion between Mn and Zn atoms. This effect is limited to the thin ZnSe capping layer (3 ML thickness), but it is expected that more Mn atoms have migrated to the surface as a result of this step. When the growth of ZnSe is resumed, Mn atoms are present on the surface of sample *c*, which gives rise to Mn segregation. In the case of sample *b*, the surface instead is mainly terminated by Se atoms and the amount of Mn atoms taking part in the segregation is very limited. Moreover, after the annealing step, the temperature drops from 600 to 300 °C. This process is relatively slow and, when the growth of the ZnSe cap is resumed, it probably proceeds at a somewhat elevated temperature (≈350 °C). The likely difference in the actual growth temperature reduces the Se sticking coefficient [26,27] and the Se/Zn ratio at the surface. Therefore, as in the case of MBE grown sample *a*, these phenomena may also enhance the Mn segregation.

## 4. Conclusions

In conclusion, the combination of APT, STM, and SIMS provides detailed information on the Mn concentration profile across a III–V/II–VI interface. Our study shows that a well defined interface between GaAs and Zn(Mn)Se is consistently created, using several growth methods. The diffusion of Mn atoms in GaAs is limited to a few nanometers from the interface of the MnSe layer, with a diffusion length λ=0.3 nm. Moreover, a substantial redistribution of Mn occurs in proximity of the MnSe/ZnSe interface, which gives rise to an intermixed Zn_1−*x*_Mn_*x*_Se layer with x=0.24. Finally, we showed that the distribution of Mn in the ZnSe layer can be controlled by the growth method and by the presence of an annealing step. Sample *a*, grown by MBE, shows a similar behavior to sample *c*, grown by MEE and annealed. In both cases, a slowly decreasing Mn concentration is retained in the ZnSe. In the case of sample *b*, which was grown by MEE, a different behavior was observed and the Mn segregation is largely suppressed. This difference in behavior can be exploited to effectively tune the distribution of Mn atoms in the ZnSe layer.

## Figures and Tables

**Figure 1 nanomaterials-10-01315-f001:**
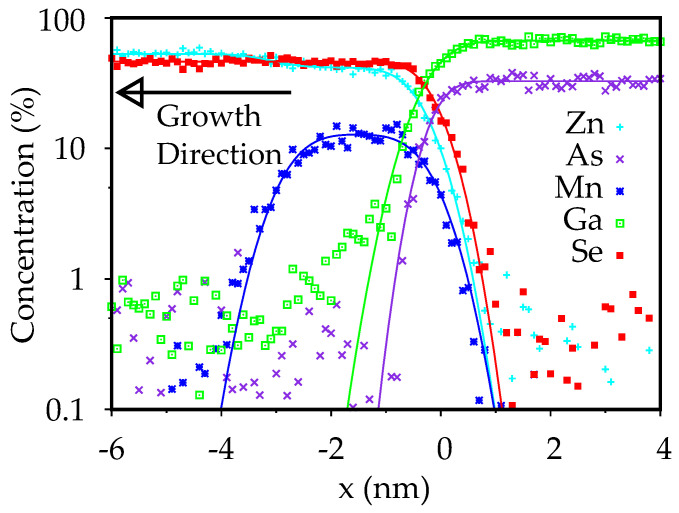
Concentration profile of each element measured by APT on sample *b*. The data are fitted using a profile based on two error functions. A good agreement between the data and the fit is obtained, and some small deviations are observed in the II–VI region for the Mn and Ga profile.

**Figure 2 nanomaterials-10-01315-f002:**
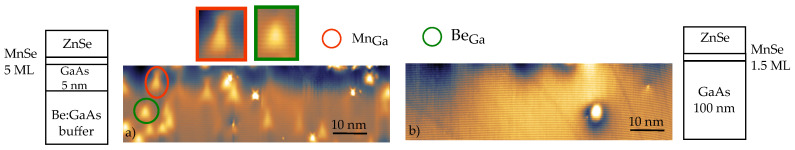
Results of the STM measurements on two different samples, both taken in empty state imaging condition (V = 1.55 V, I = 30 pA). (**a**) shows an image of the GaAs layer in sample *a*, taken about 3 nm away from the II–VI interface. Two detailed images on top of the panel show the differences in contrast between a Mn and a Be dopant; (**b**) is an image of sample *b*, taken in the intrinsic GaAs region about 10 nm away from the interface. No traces of Mn atoms are observed.

**Figure 3 nanomaterials-10-01315-f003:**
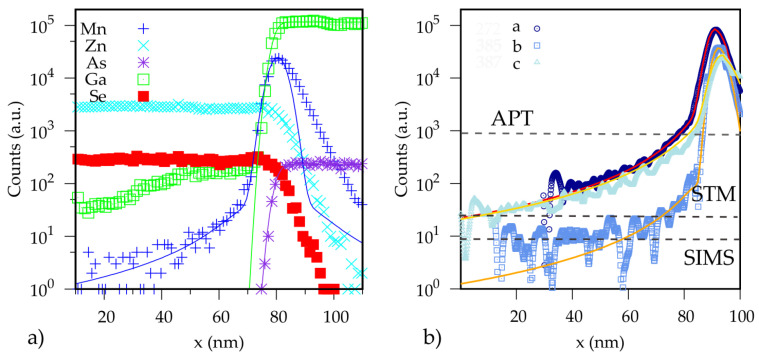
In (**a**), we show the concentration profiles measured by SIMS on sample *b*. The front tails of the profiles (Mn, Ga, As) are fitted with the fit function used for the APT measurements. A Lorentzian function is added to the Mn profile to fit the tail of the Mn distribution. In (**b**), we report the front tail of the Mn distribution for the three different samples. The noise level for each measurement technique is depicted with a dashed line.

**Table 1 nanomaterials-10-01315-t001:** Fitting parameters for the concentration profile of different elements, obtained from the APT data. The maximum element concentration (*C*), the position of the interfaces ($x_{1}$,$x_{2}$), and the diffusion length (λ) are reported.

Element	C(%)	λ (nm)	$x_{1}$ (nm)	$x_{2}$ (nm)
Mn	24±0.9	0.34±0.07	0.2±0.1	−2.84±0.1
Ga	132±1	0.34±0.05	−0.24±0.05	
As	64±1	0.35±0.08	−0.17±0.08	
Zn	104±1.7	0.33±0.06	0.34±0.07	−3.1±0.2
Se	92±0.7	0.3±0.05	0.11±0.05	

**Table 2 nanomaterials-10-01315-t002:** Fitting parameters for the concentration profile of Mn impurities, obtained with SIMS measurements. Only the front tail is considered and the profile is split in a diffusive component (*C*, λ) and a segregation component ($C_{L}$, $\lambda_{L}$).

	$C(\cdot 10^{5}\;a.u.)$	λ(nm)	$C_{L}\left(\cdot 10^{3}\;a.u.\right)$	$\lambda_{L}\;\left(\mathrm{nm}\right)$	$C_{L}/C$
*a*	1.52±0.2	1.51±0.05	10.3±0.2	4.45±0.15	0.07
*b*	0.71±0.01	1.28±0.06	0.52±0.01	7.0±0.5	0.007
*c*	0.34±0.01	1.65±0.12	5.10·±0.5	8.08±0.16	0.15

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
