# Peer review of "Design and Characterization of a Sharp GaAs/Zn(Mn)Se Heterovalent Interface: A Sub-Nanometer Scale View"

_nanomaterials, 2020, doi:10.3390/nano10071315_

Round 1

Reviewer 1 Report

This is a very nice paper describing the chemical analysis of the GaAs/ZnSe interface doped with Mn ions. Combining different highly sensitive instrumental techniques cross section scanning tunneling microscopy, atom probe tomography and secondary ions mass spectroscopy), the authors made a compiling case on the suppression of the diffusion of Mn ions in the GaAs layer with the redistribution of these ions within the ZnSe layer. As a plausible mechanism, the authors speculate that the Zn-Se bond is stronger than the Mn-Se bond, which will inhibit the segregation of Mn. Perhaps addition of numerical values of the bond energies will help the reader get a better idea about the retention of Mn in the ZnSe layer. The manuscript is well-articulated and the discussions and conclusions are well-supported by the experimental data. I therefore recommend the publication of this manuscript in Nanomaterials.

Author Response

Dear reviewer,

Thank you for taking the time checking our article and for recommending it for publication in Nanomaterial's special issue.

Kind regards,

Davide Grossi

Reviewer 2 Report

The manuscript “Design and characterization of a sharp GaAs/Zn(Mn)Se heterovalent interface: a sub-nanometre scale view” reports a very detailed study of atomic distribution across a III-V/II-VI interface, and particularly of the behavior of Mn, a magnetic impurity, across this interface. To this end the authors have employed an impressive arsenal of experimental tools (X-STM, APT and SIMS), providing important insights into the interface in question, which automatically also sheds important light on the behavior of III-V/II-VI interfaces generally.

I feel that the subject matter of the paper would be suitable for publication in Nanomaterials after the authors address the comments below.

  1. On reading the paper, I was at the outset rather misled by the extensive references to spintronic literature, that in the end turns out to be quite irrelevant. Yes, the paper does deal with issues that may be of interest to spintronic applications, but to make it so we need to know something about the magnetic properties of the systems resulting from this study. I would concede that the potential relevance to spin phenomena in semiconductors should be mentioned, and citing Ref. 1 may be OK for this purpose. However, referring to the ferromagnetic semiconductor literature (focused on GaMnAs with significant Mn content) is highly premature in the context of this paper. Additionally, it is conspicuous that the authors focus so much on III-Mn-V alloys, while not citing any serious sources from the vast literature on II-Mn-VI alloys, despite the fact that the alloy Zn(Mn)Se appears in the very title of the paper. I dwell on this relatively minor point because by reading the opening paragraphs I was initially expecting something really relevant to spintronics, like something on magnetic properties of the systems under investigation.
  2. It would be helpful to show the APT data obtained on sample C. By comparing this with the data obtained on sample B, one may learn considerably more about the effects of annealing, which is an important process affecting the properties of the interface. The data obtained by SIMS are very similar for both sample, and one cannot really distinguish the results on these two samples.  Actually, SIMS data obtained on all three samples are very similar.
  3. The y-axis label in Fig. 1 (“concentration”) is confusing. Why is the ratio of Ga:As not 1:1?
  4. The sign of x1 and x2 in Table I should be corrected. Shouldn’t x2 be negative?
  5. The authors state that Mn diffusion into GaAs is suppressed. This is clearly shown in Fig. 1 and 2. However, it is unclear why MnSe is mixed with ZnSe in the interface range.  The SIMS data suggests that both Zn and Mn are diffusive. The stoichiometry of the interface should be carefully examined, and discussed.
  6. Conspicuous by its absence in this paper is the absence of TEM images. This is a universal “staple” for discussing interfaces land could shed important light on the crystal quality of the interface that, after all, is the topic being presented.

Author Response

Dear reviewer,

We would like to thank you for your insightful comments to our paper. We carefully considered them and we would like to make the following modifications to our paper:

Point 1 : On reading the paper, I was at the outset rather misled by the extensive references to spintronic literature, that in the end turns out to be quite irrelevant. Yes, the paper does deal with issues that may be of interest to spintronic applications, but to make it so we need to know something about the magnetic properties of the systems resulting from this study. I would concede that the potential relevance to spin phenomena in semiconductors should be mentioned, and citing Ref. 1 may be OK for this purpose. However, referring to the ferromagnetic semiconductor literature (focused on GaMnAs with significant Mn content) is highly premature in the context of this paper. Additionally, it is conspicuous that the authors focus so much on III-Mn-V alloys, while not citing any serious sources from the vast literature on II-Mn-VI alloys, despite the fact that the alloy Zn(Mn)Se appears in the very title of the paper. I dwell on this relatively minor point because by reading the opening paragraphs I was initially expecting something really relevant to spintronics, like something on magnetic properties of the systems under investigation.

Response 1 : We chose to include several references to spintronic applications to provide an accurate context to our study. The realization of a sharp interface between one semiconductor doped with magnetic impurities and another with optimal opto-electronic properties is a key challenge in that field, and our aim was to indicate potential implications for such systems. We agree that the scientific literature for II-Mn-VI alloys should be better represented in the introduction of the paper, and we propose to add to the following citations :

  • J. Cibert and D. Scalbert, Diluted Magnetic Semiconductors: Basic Physics and Optical Properties, in Spin Physics in Semiconductors, ed. M.I. Dyakonov, Springer-Verlag Berlin, Heidelberg, 2008.

  • I. A. Buyanova, G. Yu. Rudko, W. M. Chen, A. A. Toropov, S. V. Sorokin, S. V. Ivanov, and P. S. Kop’ev, Control of spin functionality in ZnMnSe-based structures: Spin switching versus spin alignment, Appl. Phys. Lett. 82, 1700 (2003).

  • M. Oestreich, J. Hübner, D. Hägele, P. J. Klar, W. Heimbrodt, and W. W. Rühle,

D. E. Ashenford, B. Lunn, Spin injection into semiconductors, Appl. Phys. Lett. 74, 1251 (1999).

  • B. T. Jonker, A. T. Hanbicki, and Y. D. Park, G. Itskos, M. Furis, G. Kioseoglou, A. Petrou, X. Wei, Quantifying electrical spin injection: Component-resolved electroluminescence from spin-polarized light-emitting diodes, Appl. Phys. Lett. 79, 3098 (2001).

Point 2 : It would be helpful to show the APT data obtained on sample C. By comparing this with the data obtained on sample B, one may learn considerably more about the effects of annealing, which is an important process affecting the properties of the interface. The data obtained by SIMS are very similar for both sample, and one cannot really distinguish the results on these two samples. Actually, SIMS data obtained on all three samples are very similar.

Response 2 : We completely agree with your suggestion that APT data for sample C would have been useful to provide further proof of the effect of annealing on the interface. However the comparison between SIMS data for samples B and C shows a quite different behaviour, especially 20 nm away from the surface in the ZnSe region. On the other hand, STM measurements on sample C confirm that the back-diffusion of Mn impurities in GaAs remained extremely low, if present.

For this reason and due to the challenges in sample preparation and the involved effort of APT measurement on this type of samples we chose not to invest further resources in the investigation of multiple samples by APT.

Point 3 : The y-axis label in Fig. 1 (“concentration”) is confusing. Why is the ratio of Ga:As not 1:1?

Response 3 : This artefact is due to our choice in data analysis. The main focus of this study was the diffusion phenomena taking place at the interface between GaAs and Mn:ZnSe. In order to accurately quantify the amount of As diffusing through the interface, we decided to exclude As clusters from our analysis of the mass spectrum, which appear only in the almost pure GaAs region during APT measurements [https://doi.org/10.1016/j.ultramic.2013.02.012] due to the high As concentration. This allowed us to improve the sensitivity to As atoms in the ZnSe region, but generated an offset in the count of As atoms in the GaAs region, since the contribution of As clusters was not taken into account. As a result, an apparent difference in concentration between Ga and As was detected.

We will mention this in the improved version of our paper and explain it further in the supplementary information.

Point4 : The sign of x1 and x2 in Table I should be corrected. Shouldn’t x2 be negative?

Response 4 : Thank you for your accurate observation, we will correct the values for x2 in the table

Point 5 : The authors state that Mn diffusion into GaAs is suppressed. This is clearly shown in Fig. 1 and 2. However, it is unclear why MnSe is mixed with ZnSe in the interface range.  The SIMS data suggests that both Zn and Mn are diffusive. The stoichiometry of the interface should be carefully examined, and discussed.

Response 5 : SIMS measurements are powerful due to the high dynamic range of this techniques, but the presence of artefacts in the back tail of the concentration profile is well documented, and originates from the backscattering of ions. The absence of Mn, Zn and Se in GaAs is confirmed by both APT and STM measurements .

Point 6 : Conspicuous by its absence in this paper is the absence of TEM images. This is a universal “staple” for discussing interfaces land could shed important light on the crystal quality of the interface that, after all, is the topic being presented.

Response 6 : Using STM instead of TEM to study the cross section allowed us to observe dopants (which nearly impossible with TEM) and to get information on the type of dopants present close to the interface. STM allowed us to distinguish between different dopants found in proximity of the interface (For example in figure 2 the distinction between Be and Mn atoms). Furtheremore STM is far superior to TEM in its ability to see interface fluctuatations and structure. STM is only sensitive to a single (the surface) layer atomic layer whereas TEM will average its information of the full thickness of the lamella (typical 50 to 100 layers). For these reasons we chose for STM instead of TEM analysis.

The STM observation of an atomically flat surface in proximity of the interface after cleaving further support the high structural quality of the interface between GaAs and ZnSe. If this was not the case, a high density of surface steps would be observed in cross section.

With kind regards

Reviewer 3 Report

- Please clearly indicate the end of the y-scale in Fig. 1.
- The Ga and As concentration in Fig. 1 seem to be off the expected 50% value for both. Please clarify what is the reason for that and whether this inaccurate value can undermine the results obtained for the dopant species.
- The statement that the in-depth resolution is better than 0.1 nm sounds very optimistic. Let's assume it is better than 0.25 nm, otherwise it would be difficult to justify diffusion lengths as low as 0.3 nm. In order to ensure that the resolution of APT is below the nanometer, you should make sure that the concentration was sampled within a very narrow region around the sample axis. Is this the case?
- Comparing Fig; 1 with Fig; 3, it seems that the APT measurement indicates a Mn layer witihn the II-VI material, the SIMS measurement within the GaAs. What is the reason for that?
- The diffusion length lambda determined by APT and by SIMS differ quite significantly? What is the reason for that?

Author Response

Dear reviewer,

We would like to thank you for your insightful comments to our paper. We carefully considered them and we would like to make the following modifications to our paper:

Point 1 : Please clearly indicate the end of the y-scale in Fig. 1.

Response 1 : Thank you for your observation, we will show the plot with a y range between 0.1 and 100 %

Point 2 : The Ga and As concentration in Fig. 1 seem to be off the expected 50% value for both. Please clarify what is the reason for that and whether this inaccurate value can undermine the results obtained for the dopant species.

Response 2 : Thank you for your remark. This is a result of our data analysis that was not sufficiently explained in the manuscript. During APT of GaAs the surface becomes enriched in As (as Ga atoms are removed preferentially) and as a result a large number of As clusters are detected in the mass spectrum [https://doi.org/10.1016/j.ultramic.2013.02.012]. These clusters are not formed on the ZnSe side of the interface and in order to minimize the noise background, we excluded all clusters of As that cannot be seen in the ZnSe from the analysis.

This approach allows us to be more sensitive to As inside ZnSe and follow the diffusion profile more accurately, but an apparent difference in concentration between Ga and As in the APT results is observed.

We will mention this in the improved version of our paper and explain it further in the supplementary information

Point 3 : The statement that the in-depth resolution is better than 0.1 nm sounds very optimistic. Let's assume it is better than 0.25 nm, otherwise it would be difficult to justify diffusion lengths as low as 0.3 nm. In order to ensure that the resolution of APT is below the nanometre, you should make sure that the concentration was sampled within a very narrow region around the sample axis. Is this the case?

Response 3 : Thank you for your correction, in our study we expect an in depth resolution of about 0.2 nm, since we used a very fine z sampling around the interface (dz = 0.1 nm) and the machine inherent resolution is usually around 0.1 nm[https://doi.org/10.1063/1.3186617, https://doi.org/10.1063/1.3182351, https://doi.org/10.1046/j.1365-2818.2001.00923.x] . We will amend the text in the paper and add a description in the supplementary information on the z step used for the APT analysis.

Point4 : Comparing Fig; 1 with Fig; 3, it seems that the APT measurement indicates a Mn layer within the II-VI material, the SIMS measurement within the GaAs. What is the reason for that?

Response 4 : SIMS measurements are powerful due to the high dynamic range of this techniques, but the presence of artefacts in the back tail of the concentration profile is well documented, and originates from the backscattering of ions. The absence of Mn, Zn and Se in GaAs is confirmed by both APT and STM measurements.

Moreover the result of SIMS measurements was not normalized to display the concentration of the elements. Comparing the position of the Mn peak with the Se and As profile (which happen to have similar counts) it appears that the Mn is in an intermediate position between the Se falling edge and the As raising edge.

Point 5 : The diffusion length lambda determined by APT and by SIMS differ quite significantly? What is the reason for that?

Response 5 : The difference in in diffusion length can be attributed to the different depth resolution of APT and SIMS measurements. The typical SIMS z resolution is about 1 nm, while the APT depth resolution can be as low as 0.2 nm.

Thank you for your comments and kind regards